# Short Communication: Monitoring rock falls with the Raspberry Shakes

Andrea Manconi[1*], Velio Coviello[2], Maud Galletti[1], Reto Seifert[1]

[1]Swiss Federal Institute of Technology, Dept. of Earth Sciences, Zurich, 8092, Switzerland
[2]Free University of Bozen-Bolzano, Facoltà di Scienze e Tecnologie, Italy

*Correspondence to*: A. Manconi (andrea.manconi@erdw.ethz.ch)

**Abstract.** We evaluate the performance of the low-cost seismic sensors Raspberry Shake to identify and monitor rock fall activity in alpine environments. The test area is a slope adjacent to the Great Aletsch glacier in the Swiss Alps, i.e. the Moosfluh deep-seated instability, which is undergoing an acceleration phase since the late summer 2016. A local seismic network composed of three Raspberry Shakes was deployed starting from May 2017, in order to record rock fall activity and its relation with the progressive rock slope degradation potentially leading to a large rock-slope failure. Here we present a first assessment of the seismic data acquired from our network after a monitoring period of 1-year. A webcam was installed on the opposite side of the active slope, acquiring images every 10 minutes to validate the occurrence and identify rock falls as well as their location and approximate size. Despite seismic data were collected mainly to identify rock fall phenomena, other event types were recorded during the monitoring period. Thus, this work provides also general information on the potential use of low-cost sensors in environmental seismology.

## 1 Introduction

Rock falls constitute a major hazard in most steep natural rock slopes. The growing number of residential buildings and transport infrastructures in mountain areas has progressively increased the exposure to such processes, making the development of reliable detection systems crucial for early warning and rapid response (Stähli et al., 2015). Local geological and geomorphological conditions are the main pre-disposing factors affecting the sizes of failing rock blocks, the falling dynamics, as well as the total run out distances (Corominas et al., 2017). Different triggering agents (mainly earthquakes and/or meteo-climatic variables) have also an impact on slope failure processes, which can range from a single block fall scenario to large and more complex rock avalanches. In addition, increase of rock fall activity has been observed in areas affected by large and deep seated slope instabilities prior to catastrophic failure events (Rosser et al., 2007).

Accurate catalogues (including event location, time, and magnitude) are essential to understand and forecast rock falls, as well as other landslide processes (Kirschbaum et al., 2010). Usual approaches to build catalogues are based on chronicles and observations of past events; however, catalogues may lack of completeness, as the information is often qualitative, and constrained to limited time windows and/or specific locations. This is especially true for small- to medium-size rock fall events

(Paranunzio et al., 2016). For this reasons, there is an increasing focus on quantitative monitoring approaches, which can provide more accurate and unbiased datasets. However, continuous monitoring settings are still rare.

As rock fall phenomena induce also seismic waves (Dammeier et al., 2011; Dietze et al., 2017), seismic instruments can be installed directly on the unstable rock face to catch precursory signs of rock failure (Arosio et al., 2009), or at relatively large distances to detect a rock fall event occurrence and its propagation (Manconi et al., 2016). In particular, seismic sensors present a significant number of advantages as they are (i) compact, relatively low-cost sensors, (ii) highly adaptable to difficult field conditions, (iii) and can provide reliable information in their flat-response frequency range on a broad spectrum of mass wasting processes occurring in relatively large areas (Burtin et al., 2014; Coviello et al., 2015). Consequently, in recent years the seismic signature of rock slope failure phenomena has been investigated by several authors in different environments and monitoring set-ups (e.g., Helmstetter and Garambois, 2010; Zimmer and Sitar, 2015; Fuchs et al., 2018). The results have shown how from the seismic signals it is possible to derive information to characterize rock falls, with different level of accuracy depending upon signal sampling rates, distances between the sensors and the event, as well as the network density. High-resolution, dense seismic networks are expensive to install and need resource-intense maintenance: one high-resolution seismic station costs in the order of tens of thousands dollars to build and equip, including sensors, on-site data acquisition systems, telecommunications, and back-up power. Thus, low-cost solutions are becoming more and more attractive to increase the capability of detection and investigation of seismic activity (Cochran, 2018). Moreover, low-procurement, as well as limited installation and maintenance efforts are envisaged in case of the deployment of seismic networks including tens (or even hundreds) of sensors. In this scenario, a recently developed low-cost seismic sensors, i.e. the Raspberry Shake seismometers, provide an interesting low-cost, plug-and-play solution. The Raspberry Shake devices have become more and more popular, mainly for home use, educational purposes, and outreach. However, their potential for seismic monitoring in challenging environmental conditions is still unexplored. In this work, we show the results of a 1-year pilot test performed in the Swiss Alps, deploying a network of three Raspberry Shake seismometers to monitor rock slope failure events associated to a large, deep-seated slope instability. In the following sections, we provide a short technical description of the sensor, introduce the study area selected, and provide details on the performances of the Raspberry Shakes.

## 2 The Raspberry Shake

The Raspberry Shake (RS) is an all-in-one, plug-and-go solution for seismological applications. Developed by OSOP, S.A. in Panamá, the RS integrates the geophone sensors, 24-bit digitizers, period-extension circuits and computer into a single enclosure (see details in the Supplementary Information). Currently, available RS versions (V6+) measure ground velocities with one (1D, vertical component) or three (3D, one vertical and two horizontal components) geophones (4.5 Hz Racotech RGI-20DX) and sampling rates are adaptable up to 100 Hz. Moreover, combination of geophones with other devices like MEMS and omnidirectional pressure sensors are also available. The power supply is 5 Volts (2.5 Amp supply) and consumption is estimated in 2.8 Watts at start-up and 1.5 Watts during run-time. Data is saved on a local SD-card (default 8

Gb, but larger cards can be installed), and the estimated data amount per channel is below 10 Mb/day (~2 years of local storage). Local storage can be thus adapted depending on the SD-card mounted, the number of sensors available, and sampling rate selected. By default, time synchronization is based on NTP (Network Time Protocol), however, a GPS module can be connected via USB for situations where internet connection is not available. We refer the reader to the Supplementary Information and to the webpage https://raspberryshake.org for additional technical details on power consumption and communication issues. At the moment of our procurement (January 2017) only the RS V4 was available in the market, and thus the results and performance assessments presented here below refer to the 1D version (vertical component 4.5 Hz geophone, with 50 Hz sampling rate).

## 3 Area of study and monitoring network

The Great Aletsch Region (Swiss Alps, see Figure 1) has undergone to several cycles of glacial advancement and retreat, which have deeply affected the evolution of the surrounding landscape (Grämiger et al., 2017). In this region, the effects of the current climate change are striking, as the Aletsch glacier (blue shading in Fig. 1) is experiencing remarkable retreat with rates in the order of 50 meters every year (Jouvet et al., 2011). In particular, a deep-seated slope instability located in the southern slope of the Aletsch valley, more specifically in the area called "Moosfluh", has shown during the past decades years evidences of progressive increase of surface displacement (Kos et al., 2016; Strozzi et al., 2010). In the late summer 2016, an unusual acceleration of the Moosfluh rockslide was observed, with maximum velocities reaching locally up to 1 meter per day (Manconi et al., 2018). Such a critical evolution caused the generation of deep tensile cracks, and resulted in an increased number of rock failures at different locations of the landside body.

In this scenario, we have installed a local network composed of three RS V4 sensors. RS-1 (installed on May 19, 2017) and RS-2 (installed on June 27, 2017) are co-located within precedent monitoring infrastructures and exploit from them the necessary power (solar panels and batteries) and the internet connection (GSM) necessary for real-time data transmission (Loew et al., 2017). RS-3 (installed on July 03, 2017) is located in the basement of the Moosfluh cable-car station, and leverages from existing power and Internet connection facilities. The coupling between the station and the ground is granted through an aluminium plate (10x180x280 millimetres) screwed directly on the rock face by means of three M10 bolt anchors. The standard RS enclosure provided is made of plastic plates (5 millimetres thickness) and classified as IP10 (protected from touch by hands greater than 50 millimetres, not protected from liquids). Due to the expected harsh conditions at our monitoring locations, especially during winter periods, we assembled the RS on a IP67 (protected from total dust ingress, and from immersion between 15 centimetres and 1 meter in depth) polycarbonate enclosure (180x75x180 millimetres, model PC 175/75 HG - www.distrelec.ch) to isolate the sensor and the electronic parts from direct effects of external agents (rain, snow, wind, dust, animals, see also details in the Supplementary Information). IP67 enclosures are currently available to buy also from the RS shop (not available at the time of our procurement). Data acquired from RS-1 and RS-2 is transmitted in real-time to the ETH Zurich servers via GSM network through a mobile access router (AnyRover, see details at www.anyweb.ch). Instead, the RS-

data is stored locally and also forwarded (optional feature in the RS configuration) to a Winston Wave Server (Wave INformation STOrage Network, developed by the Alaska Volcano Observatory, to replace the Earthworm Wave Server, data resides in an open source MySQL database). RS-3 data are accessible through FDSN Web Services at the address caps.raspberryshakedata.com.

## 4 Results

### 4.1 Monitoring performance

RS-1 and RS-2 stations, both installed on the ground surface at elevations >2,000 m a.s.l. in an alpine environment, provided continuous record of seismic data since the installation without any site intervention in the 1-year monitoring period presented here. Air temperatures in this period ranged from -20 degrees in winter to +25 degrees in summer, and snow cover up to 3 meters was recorded at the RS-2 location and around 1.5 meters at RS-1 location between January and March 2018. This confirms that the enclosure we have deployed was sufficient to protect the Raspberry Pi components against alpine environmental conditions. We reported only very limited data loss (in total less than 5 minutes records over 1 year, ) at the stations RS-1 and RS-2, associated to planned system restarts after configuration changes (performed through remote access). At the RS-3 location instead, the data loss was more consistent (in total one week of data loss), due to power outage at the cable car station during a period of planned maintenance. However, the problem was unrelated to the RS-3 system itself, which started again to properly record data without intervention when the power was set back to normal. Data transfer through the cellular network links (RS-1 and RS-2) also worked smoothly during 1-year period. The results of systematic ping-tests (20 ICMP echo pings of 56 bytes every 300 seconds) show an average response time below 100 milliseconds. No remarkable network outage is reported during the period of observation (see also Supplementary Information), ensuring thus continuity for potential near-real-time analyses, as well as for the NTP service synchronization. Estimated timing quality is thus in the order of ±0.02 seconds (1 sample) or better. The current network density (3 stations with inter-station distance of about 1 km) is sufficient for detection and validation of the seismic signals but probably not enough to achieve accurate source locations. These inaccuracies can be further enhanced by time synchronization issues between the stations due to the use of NTP services; however, expected timing errors are in the order (or smaller) of the biases due to incorrect velocity models or imprecise phase picking (Lacroix and Helmstetter, 2011).

We investigated the quality of the seismic data acquired, by comparing the background noise (McNamara and Buland, 2004) of our three RS against a reference broadband seismic station (CH.FIESA, managed by the Swiss Seismological Service (SED), see details at http://stations.seismo.ethz.ch) located at about 5 km distance from the RS-1 station (Figure 2). The results show that the RS stations performed within the expected boundaries for such low-cost sensors installed at the surface (see also nominal instrumental noise levels in the Supplementary Information), in particular during the winter period. This is probably because in winter the snow cover (maximum during the observation period 3 meters at RS-1 and 1.5 meters at RS-2) protected the sensors against surficial noise sources. Moreover, during winter the glacial environment is relatively quiet compared with

the spring and summer periods, when during the day surface water runoff, as well as glacier flows, are very active and may affect the background noise levels. In addition, anthropic disturbances in this region are stronger during summer periods due to the large number of tourists visiting the Great Aletsch area. We note also how the data acquired at RS-3 systematically suffered from a higher level of noise during the cable car operational time period (between 8am and 4.30 pm local time), while during evenings and nights the background noise levels are similar to RS-1 and RS-2 stations.

## 4.2 Earthquakes

In a monitoring scenario where the main interest is to detect rock falls, recognition of earthquake events in the seismic traces is very important for two main reasons: (i) ground shaking due to local earthquakes (distances <100 km) can cause rock falls (e.g., Romeo et al., 2017), thus their identification is important to properly study the triggering factors affecting the rock slope degradation; (ii) the signals associated to distant events, such as regional earthquakes (distance >100 km) and teleseisms (distance >1000 km) have characteristics that might be similar (in terms of amplitudes and durations) with the signals caused by mass wasting phenomena (Dammeier et al., 2011; Helmstetter and Garambois, 2010; Manconi et al., 2016), and thus introduce a bias in the aimed rock fall catalogue. In order to test the performance of our local RS network, we selected seismic events from the catalogue provided by USGS (NEIC, see catalogue in the Supplementary Information, table S1), considering crustal events at depths shallower than 50 km, magnitudes larger than M2.5, and occurred at distances up to 15,000 km from our study area within 1-year time period (May 19, 2017 and May 19, 2018). We found that 47 out of the 64 selected earthquake events (~73%) were clearly visible in the waveform recorded by the RS-1 (Figure 3). As expected, the detectable magnitude as well as the signal amplitude scales with the distance from the seismic event's source. From the waveforms (Figure 3 a-d) it is possible to recognize the main differences in terms of amplitudes, duration and signal characteristics for different events.

## 4.3 Rock fall signals

About 250 rock fall events have been visually identified in the period between 1 July and 31 October 2017, using the images from the camera installed on the right side of the valley. In Figure 4 we show a selection of waveforms associated to rock fall events. Qualitative analysis on the signals recorded by the three stations may already provide preliminary indications on the rock fall processes. Considering the amplitudes and durations of the waveforms, we can derive first-order interpretations on the size of the rock fall and/or on the complexity of the event. For example, the rock fall signal recorded on 21 August, 2017 is very different from the one acquired on 19 September 2017 in terms of maximum amplitude and total duration. Indeed, the first one is associated to the failure of a single block that, however, did not run out very long due to low energy and/or unfavourable kinematic conditions (presence of obstacles, such as deep counterscarps present in the Moosfluh area, Manconi et al., 2018), while the second is associated to a relatively large rock avalanche involving several rock blocks with some of them reaching the glacier (see also pictures in the Supplementary Information). In general, the RS-2 station, which is located on the same slope affected by the rock failure at ~1 km distance from the source area, records larger amplitudes compared to

RS-1 (located in front of the rock fall area on the other side of the valley) and to RS-3 (installed a the cable car location). This is always true for the relatively small rock falls, while in case of events with longer durations (see for example the 19 September, 2017 event in Figure 4) RS-3 recorded the largest amplitude. The webcam pictures helped to confirm events recorded during day light, cloud-free conditions; however, as the majority of the events in our period of observation occurred

over night (see Supplementary Information, Figure S3), the identification is often not straightforward when there is more than one event per night. In some cases, despite clear seismic signals, we did not see any changes in the webcam pictures acquired before and after. This can be caused by lack in the pictures resolution and/or by rock fall events occurred out from the camera footprint, as well as by other processes occurring in the subsurface (i.e., creeping and stick-slip behaviour) observed also at other large rock slope instabilities (Poli et al., 2017). In Figure 5, we show a clear example where the seismic signals recorded

at the three RS stations are validated as a rock fall event by consecutive pictures. Differences in signal phases and amplitudes, as well on first arrivals, can be related to the different source-station distance, propagation of surface waves through different materials, as well as site effects at the station locations.

## 4.4 Other sources of seismic signals

We report a signal recorded on August 23, 2017 (see Supplementary Information, Figure S4) which presented typical characteristics of a surficial mass wasting, i.e. emerging onset and major spectral content between 2 and 5Hz; however, the first arrivals as well as the amplitudes were very similar at both RS-1 and RS-2 (high noise levels due to the cable car operations do not allow detect this event at RS-3). Moreover, the webcam pictures acquired before and after the event did not show changes potentially referred to a mass wasting in the local study area. Indeed, this signal is the seismic signature of the Piz

Cengalo rock avalanche (ca. 3 million cubic meters of failed material) occurred more than 100 km away from the monitoring location (Amann et al., 2018). This confirms the potential of low-cost RS sensors to detect relatively large surface mass wasting processes not only at very local scales but also at regional scales.

Apart from geophysical phenomena, we systematically observed seismic signals associated to environmental variables (such as rainfall events), of anthropic nature (for example helicopter and airplane flights) and/or of unclear source (see

Supplementary Information) during the monitoring period presented here. In the Supplementary Information we present examples of these signals. Since our future work is aimed at exploring ad-hoc algorithms to attempt automatic detection and location of the rock fall events in alpine settings, sources of disturbances on seismic signals will be carefully evaluated and further investigated to understand their nature and mitigate their effect on data analysis (Meyer et al., 2018).

## 5. Summary

In this work we show the performance of a network of three Raspberry Shakes during a 1-year pilot project aimed at testing such low-cost seismic sensors (developed for home-use) to study rock fall activity in alpine environments. Our results highlight

that, despite installation on the rock surface and only moderate protection from the expected harsh environmental conditions, the Raspberry Shake seismometers provided continuously waveforms during the 1-year observation period, without any further intervention after the installation. Continuous seismic monitoring for rock fall detection is of high relevance in alpine areas, where the use of other instruments can be hindered due to environmental conditions, logistics, and/or high/costs. We show also that low background noise levels at our Raspberry Shake stations allowed for detection of local, regional and distant earthquakes, as well as large mass wasting at relatively large distances. Currently, visual interpretation of the waveform properties in time and frequency domains allowed to discriminate between rock fall events associated with the evolution of the slope instability, e.g., rock fall phenomena of different size and run out, and seismic events, such as regional earthquakes and teleseisms. Future work is aimed at developing automatic detection and discrimination, as well as at attempting location of seismic signals due to rock falls. During the design of this pilot study, we aimed at retrieving the number of rock fall events occurred and use the event amplitudes and duration as proxy to classify their size. However, as demonstrated in this work, the performance of the Raspberry Shake in alpine environment look better than expected, and the use of higher sampling rates, as well as 3D ground velocity records instead of 1D vertical components only, might further enhance the capacity of better describing rock fall events. We thus foresee that due to their performances and low-cost, Raspberry Shakes will be more and more adopted also in research studies.

**Acknowledgments**

We thank Branden Christensen and Richard Boaz (OSOP) for detailed technical information on the RS sensors. Discussions with John Clinton (ETHZ-SED), Matteo Picozzi (University of Naples, Fedefico II), Angelo Strollo (GFZ Potsdam), and Víctor Márquez (CGEO UNAM) provided important hints during the pilot study and the paper writing. We are indebted with Robert Tanner from ETHZ-SED for the RS network communication settings and continuous support on automatic data transfer. We thank the reviewers Jan Beutel (ETHZ) and Florian Fuchs (University Vienna), as well as the editor Fabian Walter (ETHZ), for their insightful comments and suggestions to improve the manuscript.

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

# Figures

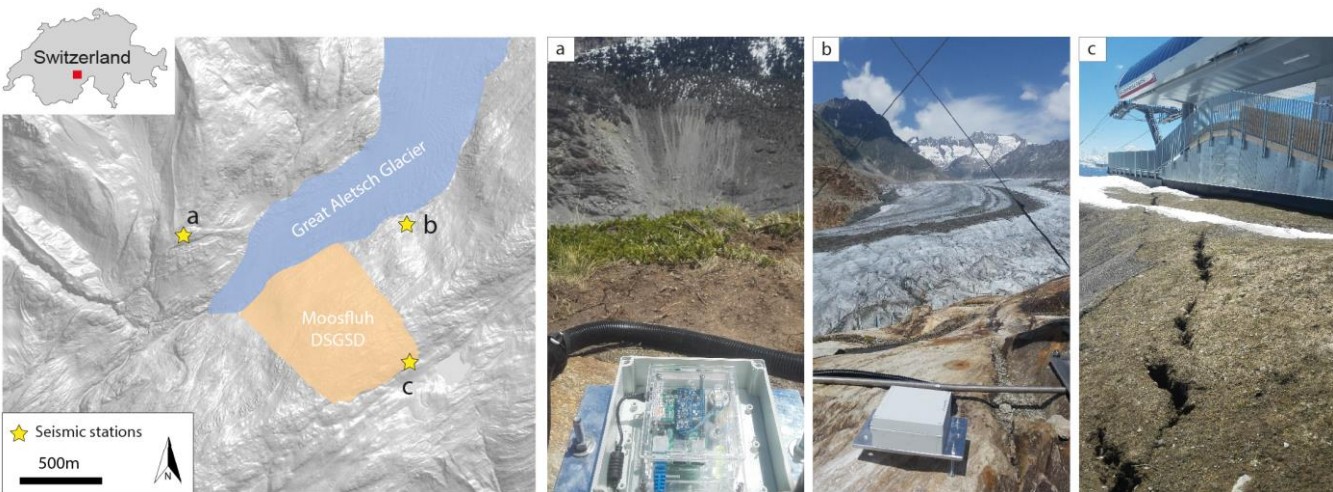

**Figure 1: (left) Map of the area of investigation with indication of the location of the three RS seismic station installed starting from May 2017. (a-c) Pictures of the RS installation (a, RS-1; b, RS-2; c, RS-3). Continuous records of seismic signals on the 3 stations are available since beginning of July 2017.**

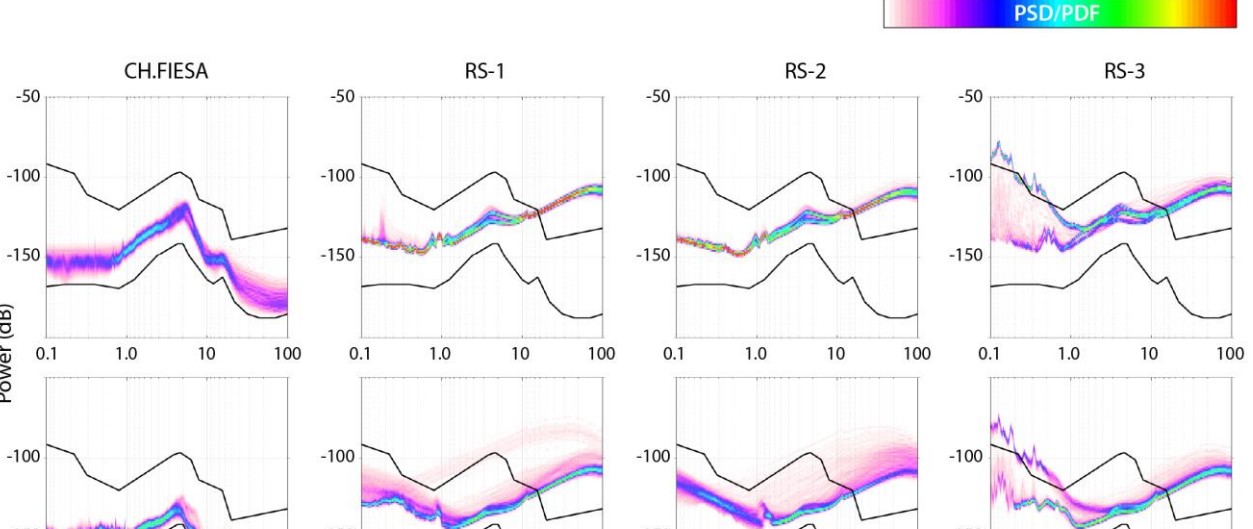

**Figure 2: Comparison of background noise levels between a broadband station (CH.FIESA) and the Raspberry Shake stations (RS-1, RS-2, and RS-3) installed in the Aletsch region for 1-year. Probability Density Functions (PDF) of the Power Spectral Densities (PSDs) were computed by stacking windows of 10 minutes in two reference weeks, one in winter (top row, 01-08 March, 2017) and one in summer (bottom row, 01-08 August, 2017). The black lines represent the Peterson's high and low reference noise models. The broadband station CH.FIESA managed by the Swiss Seismological Service is installed in the Aletsch region about 5 km away from the RS network**

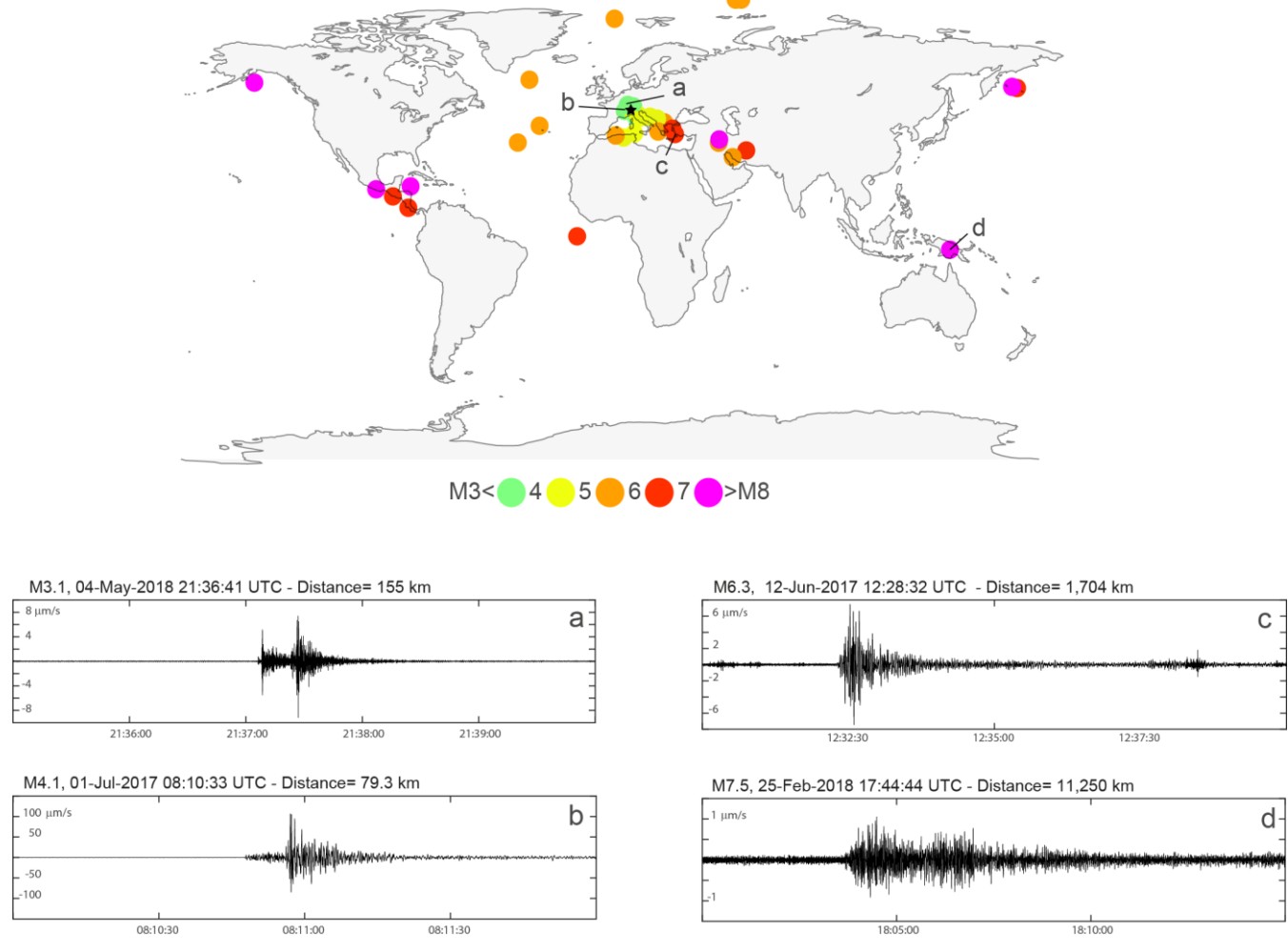

**Figure 3: Performance of the RS-1 station in recording earthquakes. (top) Spatial distribution of earthquake events identified in the RS-1 waveforms out of a catalogue of 64 earthquakes occurred within 1-year time period at distances up to 15,000 km. (a-d) Examples of seismic signal recorded by RS-1 associated to earthquakes of different magnitudes and occurred at increasing distances from the monitoring station. Signals are band-pass filtered (Butterworth) between 0.5 and 15 Hz.**

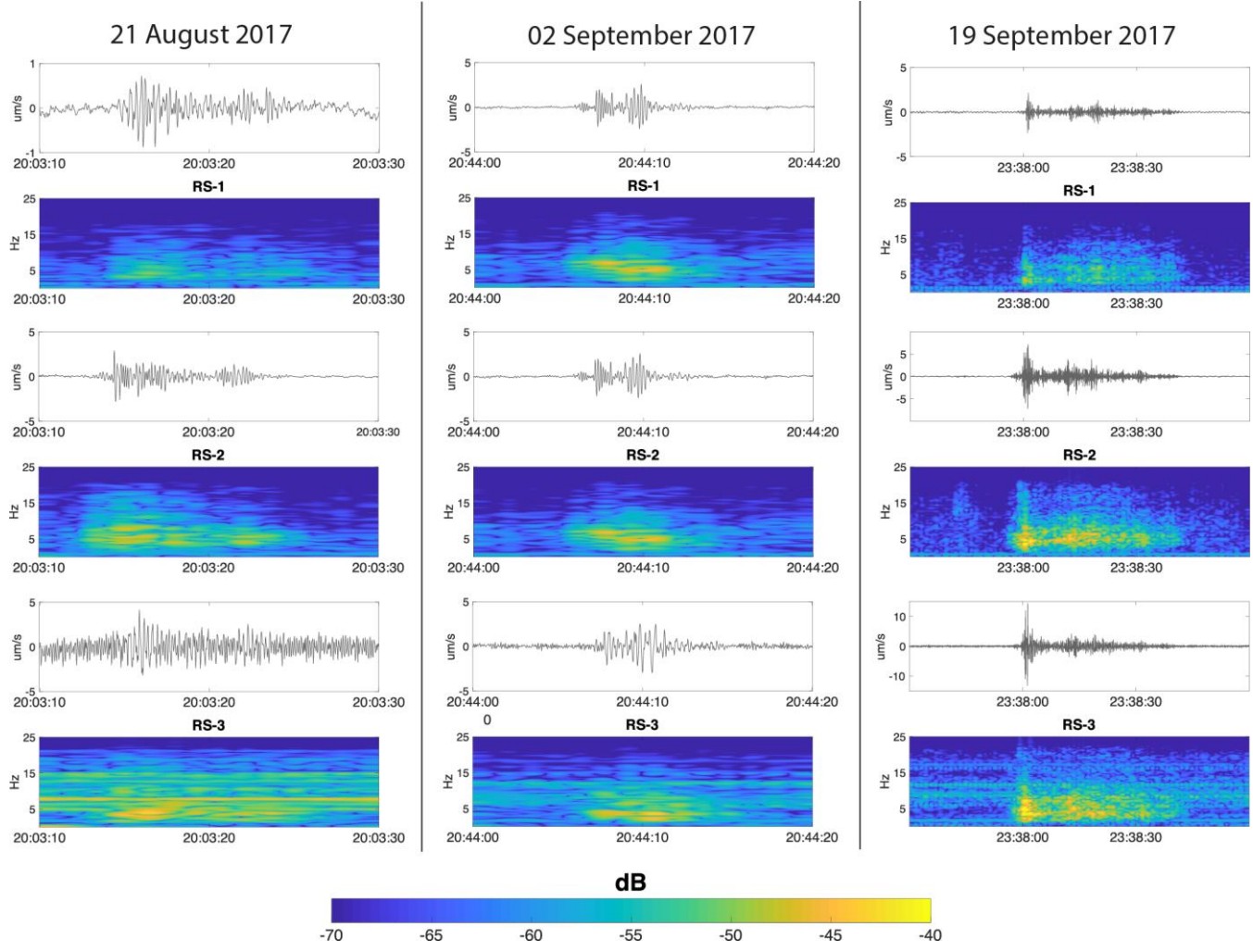

**Figure 4: Selection of signals associated to rock fall events. Signals are band-pass filtered (Butterworth) between 0.5 and 15 Hz Time is in UTC. Note the large noise level at the station RS-3 caused by the cable car operations (see also section 4 for more details).**

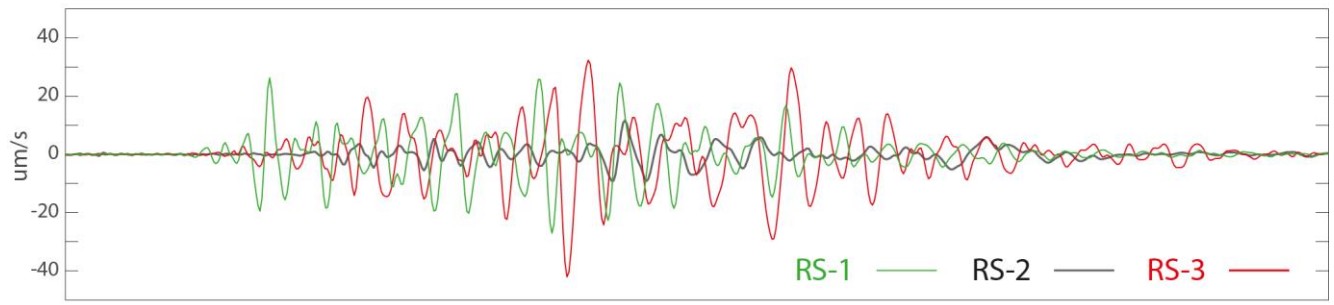

**Figure 5: Detail of a rock fall event occurred on July 27, 2017 around 15:37 UTC. (Top) Seismic signal is clearly visible at the three RS stations. Note the differences in amplitudes and phases. (Bottom) Three snapshots with 10 minutes baseline acquired by the webcam. The rock fall event is clearly visible (white circle). Future work will jointly exploit seismic and optical images to locate and characterize rock fall events.**