# Peer review of "Earth Surface Dynamics"

_Earth Surface Dynamics, 2018_

## Referee Comment (RC1) · J. Beutel (Referee) · 6 Sep 2018

**GENERAL COMMENTS**

This paper presents preliminary results of a field study using low-cost seismic sensors applied on case of a major high-alpine landslide currently developing in the Swiss Alps (Moosfluh, Valais, CH). On a secondary aspect the paper also tries to evaluate the performance of such sensors and give hints for the future use of such sensors to better quantify rock fall phenomena. I think it would be better if the authors decide on focussing on one goal for this paper, not two.

Such "cheap" seismic sensors have recently become a major attraction, not only in research but also beyond, mainly due to their accessibility (low procurement price, one-

stop-shop, simple operation) but also in the light of deploying denser seismic networks as is typically achievable with "expensive" state-of-the-art seismic sensors. Therefore a detailed and comprehensive study comparing such sensors is highly desirable and would most likely be highly cited. However the present paper does not show any comparison (to other seismic sensors) and hence cannot be seen as a "performance evaluation" as hinted in the first sentence of the abstract. Apart from the discussion of references the paper does not show any "ground truth" or "baseline" w.r.t. seismic sensors. Also, important technical details (quality of the time synchronisation, coupling of the sensors to ground, shielding from environmental influences, quantification of environmental/antropogenic influence factors) are not discussed (in great detail). Rather the paper is a preliminary experience report supported by a short presentation and qualitative alysis of data captured using three raspberry shake sensors, mainly w.r.t. earthquakes and rock fall.

The paper suggests to provide "hints" for the use of such low-cost sensors but apart from the feasibility of using such sensors on a (remote) landslide site (which in itself is not really surprising and/or novel) few hints are actually presented. A webcam for validation is frequently used in many field study - it can almost be regarded as "necessary" for current field studies. What would be really interesting to hear is how and why sensors are selected (1D vs. 3D geophone sensors, sensor characteristics), how dense a seismic network should be (given that these sensors are cheap and you suggest we can/should have more of them on a given site). How dense is dense? And what density do we need for what purpose. Also where do we need permanent networks and where can we work in campaign mode using temporary installations? Also more detailed insight into the packaging necessary, mounting on the ground and shielding from external influences would be most insightful. In high alpine settings we typically cannot bury geophone sensors under a 1m soil column or place them in a highly sheltered environment. How do you go about filtering out unwanted signals and how much of the signals aqcuired is noise? I would rather read about such aspects specific to geophones here rather than a lengthy general discussion of other monitoring methods
that could be used (top half of the second page).

Even if not feasible for you to co-locate another seismic instrument in order to compare data accross different sensor types or mounting/shielding scenarios it would be highly desirable that you discuss the difference of your system/performance vs. that of other geophone systems deployed in the aletsch moosfluh field site:

https://www.srf.ch/news/regional/bern-freiburg-wallis/die-moosfluh-kommt-nicht-zur-ruhe

https://www.tagesanzeiger.ch/panorama/vermischtes/im-wallis-droht-ein-gigantischer-bergsturz/story/26669694

Or the long-term infrastructure seismic sensors of the swiss seismological service (SED, e.g. station FIESA

http://seismo.ethz.ch/en/research-and-teaching/products-software/fdsn-web-services/

If your data is available publicly, or you have plans in providing the data set with your publication please mention and explain this.

SPECIFIC COMMENTS

Section 2 and the text in the supplementary material are identical (apart from 1-2 sentences). remove one or the other.

It is very interesting to read more details about the signal processing internals of the raspberry shake. Is this information due to personal communication or can you possibly give references for this (period extension, downsampling etc). the manuals of the raspberryshake product are rather thin here!

Supplementary material, last sentence: (Site the SEED manual). i gues you still want to include a reference here?

Figure S1 and S2 are taken (copied) from the raspberry shake product documentation:

https://manual.raspberryshake.org/specifications.html

Please cite correctly, but really what do these figures help in replication here? you can simply reference them and we look them up in the OSOP documentation.

Figure S3: You mention the period (july-october) in the text but not in the figure caption. Please also add comment/figure to the number of rockfalls observed in the other periods of your observation year. Also, is there any correlation observed to hydrological observations (rain, snowmelt)? Is the variation of rockfall events vs. hour of the day observed a real observation, or is it possibly influenced by characteristics of the instrument and changes in the environment (ie. temperature changes?)

Figure S4: It is a nice story that you detect the cengalo rockfall but without quantitative analysis it really only is a story and nothing we can really learn from here. Did you "find" this event like you described by analysis of the seismic waveform and subsequent checks with webcam images or did you specifically go and look at the time period of concern using expert/external knowledge?

Earthquake analysis results: You mention that you are able to match 47 out of 65 earthquakes in a 1Y period. Is that a good or a bad result? what is the state-of-the art? And what are the error sources? where do you see room for improvements?

Rockfall analysis results: Are you sure that your timing accuracy on the raspberry sensors over 3G radio links is good enough to perform comparisons/corellations from different sensors? I have my doubts about this knowing quite well how NTP performs over cellular links. Of course i acknowledge that in you preliminary study your analysis is mainly qualitative. But figure 4 talks about phase differences and if your network timping is only off by a few milliseconds this data will look very different!

Why do you choose a 1D rasperry shake and not the 3D or 4D (1-axis geophone, 3-axis accelerometers) models?

this may be a recent and interesting piece to look at:

https://www.earth-surf-dynam-discuss.net/esurf-2018-60

**ESurfD**

Interactive
comment

---

## Referee Comment (RC2) · F. Fuchs (Referee) · 14 Sep 2018

Review of Monitoring rock falls with the Raspberry Shakes by A. Manconi et al

The manuscript aims at providing insight into the usability of low-cost seismic sensors for purposes of environmental seismology under harsh environmental conditions. Specifically, the authors want to report on the use of Raspberry Shake devices for rockfall monitoring in the high Alps.

Recently, there is increased interest among the community in so called large-N installations with low cost sensors. Specifically for targeting high-frequency seismic radiation of local seismic sources such sensors potentially offer good monitoring capabilities for low budget. The Raspberry Shake devices have recently become more popular, however mainly for educational purposes and outreach. Thus, a study documenting their applicability under challenging climatic conditions such as high-mountain areas would be very welcome.

Unfortunately, the manuscript falls short of it's promises and lacks a number of interesting topics which could be addressed.

Short-period sensors are already commonly used in seismic monitoring of rockfalls in various environments. Thus, the sensor technology of Raspberry Shake is not new and we know that geophone based sensors are well suited to record the high-frequency seismic radiation of near-surface processes such as rockfalls. However, the Raspberry Shake is designed as an ready-to-use all-in-one solution mainly for home-use. It would certainly be interesting to read how this design is suitable for operation in high-mountain areas. As far as I know there are no other studies reporting on this. Unfortunately, especially this part is not properly addressed in the manuscript. I would expect that after one year of operation, the authors gained quite some insight into data quality, system reliability and unique challenges and how to solve them. This would certainly be interesting to read. Please see more detailed comments about this below.

The manuscript is also very brief and simplistic on the description of methods to detect events. It is nice to see that both regional and teleseismic earthquakes are well-recorded and that visually confirmed rockfalls are as well detected. Still, this is somewhat no surprise. In order to really judge the performance of such a network for rockfall monitoring purposes, much more details on the methods should be provided and challenges in data analysis should be highlighted or at least mentioned. Especially, a section about location is missing. Below I list several suggestions for improvement.

I do realize that the authors designed their work as "Short Communication" and that several of my suggestions would require additional work and consequently the manuscript would become longer. Thus I encourage the authors to spend some more time on the manuscript and shape it into a more conclusive study about the performance of Raspberry Shakes in mountain areas. If the authors prefer to keep it brief and not elaborate in more detail about the data analysis challenges, still I suggest to improve those parts that really concern the specific challenges of operating Raspberry Shakes in alpine conditions: installation, system reliability, data quality.

Best regards, Florian Fuchs
* * *
Detailed comments:

Section 1

- The introduction is somewhat lengthy and introduces general considerations not directly related to the topic of "Raspberry Shakes for rockfalls". Maybe e.g. some parts about remote sensing could be dropped to keep it short.

Section 2

- This section is in part both too brief and too extensive. Some information is overly detailed (e.g. bits for dynamic range, analog chain of processing, resistivity of Geophone, A/D sampling, ...). I suggest to refer to the manufacturer's spec sheet for such details. Stating that it's a 4.5 Hz geophone with eigenperiod electronically extended to 2s should do the job.

- On the other hand I miss information about e.g. range of available models (1 comp., 3 comp, pressure sensors, etc.), available housings, available connections. Are sampling rates adjustable? How is time synchronized? What's the power consumption? How is data stored? As this might be the first work introducing Raspberry Shakes for scientific field deployments, such details should be addressed.

Section 3

- Have there been other geophysical studies in the same area before? What other monitoring facilities are in place? The most interesting part is how to install a Raspberry

Shake in alpine conditions for long-term use. Please slightly expand your descriptions here. How is the Raspberry Shake fixed to the ground? Do you put additional shielding against wind or thermal effects? How is time synchronized? Photographs showing more details would be of great help – maybe as supplemental material.

Section X

At this part of the manuscript I am missing a section about the performance of the network. What's the amount of data retrieved? Where there any power downs? Why so? What's the quality of the data? Any gaps? What's the noise level? I would certainly like to see some PPSD/PDFs showing the noise levels over one year. Is the relative timing good enough? Any other issues due to environmental conditions?

Section 4

- Generally, this section is very simple and should be greatly supported by a description of the methods you used. My general suggestions is: instead of elaborating on individual events that after all are hard to interpret, I suggest to make use of the long observation time and large number of detected events. Are there some patterns to be seen? How do amplitudes distribute? Is it in fact possible to locate the events? Is all of this actually possible with your dataset? If yes – please show! If no – try to discuss limitations.

- The detection threshold of the network will rely on the average noise levels throughout the year. Can you give an estimate of what's the smallest ground motion you can reliably detect? This would be of relevance for detecting both earthquakes and rockfalls. What are the settings for the STA/LTA detector?

- How did you identify the 250 events as rockfalls? This is a very crucial task for any monitoring so please be more specific on how you identify those. Did you also use an STA/LTA? On how many stations do you see those events? How many of the events are confirmed by the webcam? Was it a visual inspection of the seismic data or some

automatic processing? If the latter, please provide parameters or criteria.

- How many events did you detect in total, how many of those are of "unknown" type? How did you e.g. identify rainfall (example in supplemental material) in your seismic data? How the helicopter?

- Figure S3 is quite an interesting observation! Obviously one needs to trust the detections (thus, see comment above), but certainly this is a nice observation that should be moved into the main body of the manuscript.

- Unfortunately the webcam images (Figure 3) are very hard to interpret (especially when printed on paper). Are these the best examples you can show? What's your experience with the 10 minute sampling rate? Is it sufficient? What would be the effort to increase it?

- I am missing a paragraph dedicated to locating the detected rockfalls. After all this is one of the most important tasks for monitoring purposes. Did you attempt to locate the events? How could it be done? How well does it work? Are there specific drawbacks?

- You try to qualitatively interpret some of the amplitude readings for individual events. In general it would be interesting if the recorded amplitudes relate to e.g. distance or ground type underneath the stations or between the stations and the rockfalls. Do you see any general behavior? With 250 rockfalls you should have quite a nice database to dig into this. Is there maybe an effect due to ground-coupling of the sensors?

- You show several examples of seismically recorded rockfalls, yet your description of those events is very basic. In fact, it is no surprise that you are able to detect rockfalls in the close vicinity with the Raspberry Shake. Yet, already from looking your examples it is somewhat obvious (and known from other studies) that rockfall seismic signals are quite complex and relating them to rockfall mechanisms in real field applications is not necessarily straight forward. The real challenge of any monitoring system lies in safely classifying the rockfall events and to discriminate them from all the wealth of other

ESurfD
seismic signatures (e.g. anthropogenic) in your data. This starts at classifying seismic signal as "rockfalls" and continues into scaling relations between seismic observables (e.g. amplitude, frequencies, energy, . . .) and rockfall parameters (volume, runout, mechanism, . . .). I think there is interesting work in that direction to be done with this dataset, but it's likely out of scope for the current manuscript. You may decide to work into that direction, but for the moment it would maybe be already sufficient if you add a section that discusses these challenges.

———————————————————————————

Specific remarks/typos/grammar:

Page 1, Line 16: remove "investigations"

Page 1, Line 24: remove "of"

Page 1, Line 30: what do you mean by arbitrary information?

Page 2, Line 3: replace "and" with "to"

Page 2, Line 5: replace "Despite" with "However"

Page 2, Line 26: ...from the seismic signal IT is possible . . .

Page 3, Line 1: Explain IoT

Page 3, Lines 3ff: state somewhere that the sensors measure ground velocity

Page 3, Line 18: remove "to"

Page 4, Line 1: change to: . . . isolation from different external effects (... )

Page 4, Line 6: what is a Winston Wave Server? What is it good for?

Page 4, Line 24: replace "lower" with "shallower"

Page 5, Line 5: Figures 3 → Figure 3

Page 5, Line 9: add reference to Figure

Page 5, Line 14: replace "presents" with "measures" or "records"

Page 5, Line 29: replace "coeval" with "consecutive"

Page 5, Lines 26ff: This part should be moved upwards, since it contains the detection/verification of events

Page 5, Line 31: Give examples for "other processes occuring in the subsurface"

Page 6, Line 1: Change to "Apart from rock fall phenomena . . . signals associated to other environmental processes (such as. . .)

Page 6, Lines 11ff: You should only summarize your findings. The advertisement for low cost solutions should be moved to the introduction section.
* * *
Figure 1:

- Add arrow indicating the line-of-sight of the webcam

- DSGSD is not explained

- Show inside installation of RS-3 as well

Figure 2:

- Convert Lat/Lon to distance or add distance to the plot titles.

- Caption: b) Your plots show examples of detected events and maybe estimate a lower threshold. There is no information about the real probability of detecting an event.

Figure 3:

- Any chance to improve the contrast/visibility of the webcam pictures? It is really hard to see any changes in the before/after pictures. Why not show some examples from

daylight period? These should show a clear before/after signature?

- Caption: august → August

Figure 4:

- Please describe this Figure in more detail in the manuscript text. You show the waveforms in great detail. If there is something noteworthy to be seen, please state so.

- What's the filter applied to the waveform data?
* * *
Supplemental material:

- The text is mainly a copy of Section 2 and I think it can be dropped. If you like to describe the very specific details of the signal processing inside the Raspberry Shake, you could do that here (and not in Section 2).

- Figures S1 and S2 are copied from the producer spec sheet. You could just refer to that sheet. If you like to keep them, please indicate the source.

- If you have any, here would be the perfect place to put more examples (waveforms + webcam pictures) of rockfalls (and maybe other sources: cable car, mountaineers, ...)

---

## Author Comment (AC1) · 24 Sep 2018

We thank the reviewers or the comments and constructive criticism (it is very much appreciated you both decided to do that openly and not anonymously). From the comments it is clear to us that the work presented is of interest and may be suitable for the Esurf audience after addressing the reviewers' comments. As outlined by F. Fuchs, the initial idea of this contribution was a short communication to share with the community our experience done with the Raspberry Shakes (RS) in the alpine environment. We agree that a direct comparison (or "baseline", as mentioned by J. Beutel) w.r.t. other seismic sensors is now not present and will be very helpful to better assess the RS performances. We will work in this direction to include these aspects in the revised version of the manuscript. In addition, we will consider all the points raised by the

reviewers to provide more information in the direction "performance assessment" and use of RS in alpine environments. However, an in-depth description of the detection algorithms as well as of the rock fall catalogue obtained with the RS network is out of the scope of this work, because this is the focus of our current research activities and will be included in future publications. By following the reviewers' instructions the manuscript size (in terms of text and figures) will surely increase, and we thus kindly ask to the editors to advice on whether we should keep the format of our contribution as "Short Communication" or change it towards a standard "Research Article". This will be extremely helpful to better calibrate the revision process.

Kind regards (Andrea Manconi and co-authors)

---

## Referee Comment (RC3) · F. Fuchs (Referee) · 28 Sep 2018

Dear Authors, dear Andrea, dear Editor,

first of all, I wasn't able to find the ESurf criteria for "Short Communication" compared to "Research Article" so I'm not sure where the limits in terms of Figures/Pages and requirements in terms of scientific impact are ... I also think it doesn't matter too much.

I think putting more emphasize on the instrumental part (as suggested in the reviews) and the performance of the sensors in such environments would already significantly improve the manuscript, while still allowing to keep it rather short.

In my view it's alright if you decide to leave a detailed discussion on detection algorithm / event classification / rockfall catalog for future work (there's a lot one could do and it

will take time). Anyways, I believe some of the points raised in the review can be answered at least briefly without overly expanding the manuscript (e.g. briefly outline how the 250 events database was compiled, state some criteria, etc.).

You don't have to rush it, I think, so if switching to "Research Article" gives you more time and space - go for it.

If you still prefer "Short Communication" but are running out of space: a very drastic solution (but somewhat OK in my view) would be to drop the example events section (4.1 - 4.3) completely, or drastically shorten it, or move it in great parts to Supplemental Material (e.g. show example records in Supplemental, without describing in greater detail).

This is my personal view, I'm not sure if the others would agree.

Best regards, Florian Fuchs

---

## Author Comment (AC2) · 4 Oct 2018

Dear Jan, thanks for your detailed comments to our manuscript.

As suggested, in the revised version we will better calibrate the target of our contribution, focusing more on preliminary results and performance evaluation of the Raspberry Shake (RS) at Moosfluh. Regarding the internal processing chain of the RS, we will include some more details we are aware of but some information is not available to us (we already asked) because it is OSOP proprietary information. However, we will also include a comparison to a reference seismic station located in the vicinity of our network (i.e., FIESA) to corroborate data and results gathered with the RS. Then, we will remove overlapping technical parts present both in Section 2 and in the supple-

mentary material. About your question "Why do you choose a 1D RS and not the 3D or 4D", when we have acquired and installed the RS at Moosfluh the versions 3D and 4D were still not available. The performance analysis of earthquake detection presented in Figure 2 is based on the STA/LTA parameters used and on the travel-time model considered. In any case, the "baseline" would be 100% detection; we will better clarify this point in the revised version. Concerning the analysis of the rockfall catalog, we agree on the potential effect of NTP on timing accuracy. The best timing accuracy we can get with the RS is within 1 sample, thus 0.02 seconds in our case (as per tech. spec.). As far as we know, NTP inaccuracies are usually associated to network outage, and we are investigating this point specifically. We will plot Figure 4 by showing the potential effects of time inaccuracies, showing that this will not affect the differences in amplitude between the three stations and few milliseconds will produce only a minor change in the phase. The only problem we foresee for inaccurate timing is while attempting source location. However, we speculate that the timing error in our case will be of the same order (or even smaller) than the location errors produced by considering an inaccurate velocity model. The details about the NTP effects will be clarified in our revision. Since you mention to know quite well how NTP performs over cellular links, we would appreciate if you could specifically indicate some reference or details that can help us! In the supplementary material, we will cite correctly the OSOP documentation but keep figures S1 and S3 because we think that not all potential readers of the manuscript will go and check the RS specifications and manuals, thus we think is useful to have it also here. Figure S3: we will include the information on the investigated period of time (July-October) also in the figure caption. Regarding the correlation with environmental/climatic variables, this is currently under investigation; however, there is not an evident effect due to the instrument used. Figure S4: this event was found manually; we wanted to show that the sensitivity of these instruments is good not only for local mass movements but also to detect landslides occurring far from the installation. The point will be further clarified in the revised version.

Best regards, Andrea Manconi and co-authors

**ESurfD**

Interactive
comment

---

## Author Comment (AC3) · 4 Oct 2018

Dear Florian, thanks for your careful reading of our manuscript. Please find our initial replies to your criticism.

In the revised paper, we will focus more on monitoring results and performance evaluation of the Raspberry Shake (RS) at Moosfluh, better discussing how the low-cost, all-in-one solution design of RS is suitable for operation in high-mountain areas. In particular, as suggested, we will present specific challenges of using the RS in alpine environment concerning installation, system reliability, and data quality assessment. The methods of automatic detection and location are the focus of our current research. In this paper, we want to focus more on the performances of the instruments in alpine

[Figure]

**ESurfD**
* * *
environment and we will better outline this part in the revised version. We will revise the introduction, skipping general considerations not related to the topic and reducing some parts about remote sensing. Concerning Section 2, we will move all technical details (bits for dynamic range, analog chain of processing, resistivity of Geophone, A/D sampling) to the supplementary material, where we will also include some information on the range of available RS models, power consumption, time synchronization. Section 3: we will expand the site description presenting other studies and monitoring activities performed in the same area. We will present a comparison of PPSD of the RS for different time periods (winter vs summer) against a broadband station (CH.FIESA). Section 4: as already outlined in a previous reply, most of the additional information required is part of our current research activity and is not the focus of this manuscript. Thanks for all specific remarks, we will address these in the revised version.

Best regards, Andrea Manconi and coauthors

---

## Author Response (AR1)

J. Beutel (Referee)

janbeutel@ethz.ch

**Dear Jan, thanks for your detailed comments. Please find below our initial reply to you criticism.**

GENERAL COMMENTS

This paper presents preliminary results of a field study using low-cost seismic sensors applied on case of a major high-alpine landslide currently developing in the Swiss Alps (Moosfluh, Valais, CH). On a secondary aspect the paper also tries to evaluate the performance of such sensors and give hints for the future use of such sensors to better quantify rock fall phenomena. I think it would be better if the authors decide on focussing on one goal for this paper, not two.

**Reply: Thanks for this comment, in the revised version we recalibrate the target of our contribution, focusing more on preliminary results and performance evaluation at Moosfluh test-site.**

Such "cheap" seismic sensors have recently become a major attraction, not only in research but also beyond, mainly due to their accessibility (low procurement price, one-stop-shop, simple operation) but also in the light of deploying denser seismic networks as is typically achievable with "expensive" state-of-the-art seismic sensors. Therefore a detailed and comprehensive study comparing such sensors is highly desirable and would most likely be highly cited. However the present paper does not show any comparison (to other seismic sensors) and hence cannot be seen as a "performance evaluation" as hinted in the first sentence of the abstract. Apart from the discussion of references the paper does not show any "ground truth" or "baseline" w.r.t. seismic sensors. Also, important technical details (quality of the time synchronisation, coupling of the sensors to ground, shielding from environmental influences, quantification of environmental/antropogenic influence factors) are not discussed (in great detail). Rather the paper is a preliminary experience report supported by a short presentation and qualitative alysis of data captured using three raspberry shake sensors, mainly w.r.t. earthquakes and rock fall.

The paper suggests to provide "hints" for the use of such low-cost sensors but apart from the feasibility of using such sensors on a (remote) landslide site (which in itself is not really surprising and/or novel) few hints are actually presented. A webcam for validation is frequently used in many field study - it can almost be regarded as "necessary" for current field studies.

**Reply: concerning the performance evaluation, in the revised version of our manuscript we included a comparison of RS data with waveforms gathered at the broadband station (see section 4.1). We agree, the installation of a monitoring camera is a rather standard nowadays, however, systematic validations of the rockfalls detected from seismic sensors with a camera sampling every 10min are (to the best of our knowledge) very rare.**

What would be really interesting to hear is how and why sensors are selected (1D vs. 3D geophone sensors, sensor characteristics), how dense a seismic network should be (given that these sensors are cheap and

you suggest we can/should have more of them on a given site). How dense is dense? And what density do we need for what purpose. Also where do we need permanent networks and where can we work in campaign mode using temporary installations?

**Reply: for the specific purpose described in our work (rock fall detection) 1D geophones are enough. The issue about density is dependent on the goal, on the size of the area of interest, and on the size of expected rock falls, and of course on the available budget. A full analysis of these aspects is, however, beyond the scope of our manuscript. Looking at our specific purpose (rock fall detection), one station can be enough, two are recommended to avoid false alarms, the third one is for a minimal redundancy. Continuous monitoring is envisaged to analyze the slope activity and its relationship to other environmental variables. We have highlighted these points at section 5 of the revised manuscript.**

Also more detailed insight into the packaging necessary, mounting on the ground and shielding from external influences would be most insightful. In high alpine settings we typically cannot burry geophone sensors under a 1m soil column or place them in a highly shelterd environment. How do you go about filtering out unwanted signals and how much of the signals aqcuired is noise?

**Reply: We have now added more information in section 3 and in the Supplementary Information**

I would rather read about such aspects specific to geophones here rather than a lengthy general discussion of other monitoring methods that could be used (top half of the second page). Even if not feasible for you to co-locate another seismic instrument in order to compare data accross different sensor types or mounting/shielding scenarios it would be highly desirable that you discuss the difference of your system/performance vs. that of other geophone systems deployed in the aletsch moosfluh field site:

https://www.srf.ch/news/regional/bern-freiburg-wallis/die-moosfluh-kommt-nicht-zurruhe

https://www.tagesanzeiger.ch/panorama/vermischtes/im-wallis-droht-ein-gigantischerbergsturz/

story/26669694

Or the long-term infrastructure seismic sensors of the swiss seismological service

(SED, e.g. station FIESA

http://seismo.ethz.ch/en/research-and-teaching/products-software/fdsn-web-services/

If your data is available publicly, or you have plans in providing the data set with your publication please mention and explain this.

**Reply: Agreed, we have deleted general discussion on other monitoring methods and now included a comparison to a reference SED broadband station located in the vicinity of our network (i.e., CH.FIESA). The other seismic data mentioned are not available to us. The data acquired at our station RS-3 is publicly available.**

SPECIFIC COMMENTS

Section 2 and the text in the supplementary material are identical (apart from 1-2 sentences). remove one or the other. It is very interesting to read more details about the signal processing internals of the raspberry shake. Is this information due to personal communication or can you possibly give references

for this (period extension, downsampling etc). the manuals of the raspberryshake product are rather thin here!

**Reply: overlapping parts have been removed. Regarding the details on internal processing of the raspberry shakes, this is not available to us because it is OSOP proprietary information**

Supplementary material, last sentence: (Site the SEED manual). i gues you still want to include a reference here?

**Reply: Solved. Thanks for pointing out.**

Figure S1 and S2 are taken (copied) from the raspberry shake product documentation:

https://manual.raspberryshake.org/specifications.html

Please cite correctly, but really what do these figures help in replication here? you can simply reference them and we look them up in the OSOP documentation.

**Reply: We cited correctly the OSOP documentation but keep the figures also here (agreed by OSOP, personal communication). Not all potential readers of the manuscript will go and check the Raspberry Shakes specifications and manuals, thus we think is useful to have it also here.**

Figure S3: You mention the period (july-october) in the text but not in the figure caption. Please also add comment/figure to the number of rockfalls observed in the other periods of your observation year. Also, is there any correlation observed to hydrological observations (rain, snowmelt)? Is the variation of rockfall events vs. hour of the day observed a real observation, or is it possibly influenced by characteristics of the instrument and changes in the environment (ie. temperature changes?)

**Reply: We included the period also in the figure caption. The other periods are currently under investigation. Regarding the correlation with environmental/climatic variables, this is currently under investigation; however, there is not an evident effect of influences due to the instrument used.**

Figure S4: It is a nice story that you detect the cengalo rockfall but without quantitative analysis it really only is a story and nothing we can really learn from here. Did you "find" this event like you described by analysis of the seismic waveform and subsequent checks with webcam images or did you specifically go and look at the time period of concern using expert/external knowledge?

**Reply: This event was identified visually, as the others presented in this work. We want to show that the sensitivity of these instruments is good not only for earthquakes or very local mass movements but also to detect relatively large landslides occurring far from the installation site.**

Earthquake analysis results: You mention that you are able to match 47 out of 65 earthquakes in a 1Y period. Is that a good or a bad result? what is the state-of-the art? And what are the error sources? where do you see room for improvements?

**Reply: In the revised version we opted for a manual check of the (actually) 64 selected events, as the best STA/LTA parameters for the detection of both rockfalls and earthquakes are still under study. The visual analysis confirmed that the 47 events are clearly identified in the seismic records, while the rest of events are barely visible. Due to the magnitude and distances selected for this test, the expectation was to identify 100% of the events.**

Rockfall analysis results: Are you sure that your timing accuracy on the raspberry sensors over 3G radio links is good enough to perform comparisons/corellations from different sensors? I have my doubts about this knowing quite well how NTP performs over cellular links. Of course i acknowledge that in you preliminary study your analysis is mainly qualitative. But figure 4 talks about phase differences and if your network timping is only off by a few milliseconds this data will look very different!

**Reply: We have investigated the potential effect of NTP on timing accuracy. The advertised timing accuracy for the RS is within 1 sample, thus 0.02 seconds in our case (as per tech. spec.). NTP inaccuracies are associated to network outage. In the Supplementary information we have included some detail on the NTP approach as well as the results of ping tests during the 1 year of observation for two stations (connected via 3G). As the network connections was very stable during our observation, we assume that the accuracy is in the order of 0.02 seconds or better. We attempted representing this deviation in the figure 5, however, inaccuracy would be of the same order of the line thickness of the current plot, thus no evident deviations on the phases would be noticeable.**

**The only problem we foresee for inaccurate timing is while attempting source location. However, the timing error at our scale of observation will be more affected by errors by considering intrinsic errors of homogeneous velocity models used for standard location. We refer to the RS website for further details regarding the timing accuracy.**

Why do you choose a 1D rasperry shake and not the 3D or 4D (1-axis geophone, 3-axis accelerometers) models?

**Reply: when we have acquired and installed the RS, the versions 3D and 4D were still not available.**

this may be a recent and interesting piece to look at: https://www.earth-surf-dynam-discuss.net/esurf-2018-60

**Reply: Thanks, we have now included this as reference in our section 4.**

F. Fuchs (Referee)

florian.fuchs@univie.ac.at

**Dear Florian, thanks for your detailed comments. Please find our initial replies to your criticism.**

Review of Monitoring rock falls with the Raspberry Shakes by A. Manconi et al

The manuscript aims at providing insight into the usability of low-cost seismic sensors for purposes of environmental seismology under harsh environmental conditions. Specifically, the authors want to report on the use of Raspberry Shake devices for rockfall monitoring in the high Alps. Recently, there is increased interest among the community in so called large-N installations with low cost sensors. Specifically for targeting high-frequency seismic radiation of local seismic sources such sensors potentially offer good monitoring capabilities for low budget. The Raspberry Shake devices have recently become more popular, however mainly for educational purposes and outreach. Thus, a study documenting their applicability under challenging climatic conditions such as high-mountain areas would be very welcome.

**Reply: Thanks, this shows that our study might be of interest for the community.**

Unfortunately, the manuscript falls short of it's promises and lacks a number of interesting topics which could be addressed. Short-period sensors are already commonly used in seismic monitoring of rockfalls in various environments. Thus, the sensor technology of Raspberry Shake is not new and we know that geophone based sensors are well suited to record the high-frequency seismic radiation of near-surface processes such as rockfalls. However, the Raspberry Shake is designed as an ready-to-use all-in-one solution mainly for home-use. It would certainly be interesting to read how this design is suitable for operation in high-mountain areas. As far as I know there are no other studies reporting on this. Unfortunately, especially this part is not properly addressed in the manuscript. I would expect that after one year of operation, the authors gained quite some insight into data quality, system reliability and unique challenges and how to solve them. This would certainly be interesting to read. Please see more detailed comments about this below.

**Reply: We have now added more information to this technical point (sections 2 and 4.1).**

The manuscript is also very brief and simplistic on the description of methods to detect events. It is nice to see that both regional and teleseismic earthquakes are well recorded and that visually confirmed rockfalls are as well detected. Still, this is somewhat no surprise. In order to really judge the performance of such a network for rockfall monitoring purposes, much more details on the methods should be provided and challenges in data analysis should be highlighted or at least mentioned. Especially, a section about location is missing.

**Reply: The methods of automatic detection and location are currently focus of our research. In this paper we want to focus more on the performances of the instruments in alpine environment, while leaving more detailed analyses on the dataset for future studies. We have now mentioned challenges regarding location in the section 5.**

Below I list several suggestions for improvement. I do realize that the authors designed their work as "Short Communication" and that several of my suggestions would require additional work and consequently the manuscript would become longer. Thus I encourage the authors to spend some more time on the

manuscript and shape it into a more conclusive study about the performance of Raspberry Shakes in mountain areas. If the authors prefer to keep it brief and not elaborate in more detail about the data analysis challenges, still I suggest to improve those parts that really concern the specific challenges of operating Raspberry Shakes in alpine conditions: installation, system reliability, data quality.

**Reply: We followed this second point suggested, i.e. specific challenges of using the RS in alpine environment concerning installation, system reliability, and data quality assessment.**

Best regards, Florian Fuchs

————————————————

Detailed comments:

Section 1

- The introduction is somewhat lengthy and introduces general considerations not directly related to the topic of "Raspberry Shakes for rockfalls". Maybe e.g. some parts about remote sensing could be dropped to keep it short.

**Reply: Agreed.  We have now deleted the unnecessary parts.**

Section 2

- This section is in part both too brief and too extensive. Some information is overly detailed (e.g. bits for dynamic range, analog chain of processing, resistivity of Geophone, A/D sampling, ...). I suggest to refer to the manufacturer's spec sheet for such details. Stating that it's a 4.5 Hz geophone with eigenperiod electronically extended to 2s should do the job.

- On the other hand I miss information about e.g. range of available models (1 comp., 3 comp, pressure sensors, etc.), available housings, available connections. Are sampling rates adjustable? How is time synchronized? What's the power consumption? How is data stored? As this might be the first work introducing Raspberry Shakes for scientific field deployments, such details should be addressed.

**Reply: All the required information has been now added section 2 and in the Supplementary material.**

Section 3

- Have there been other geophysical studies in the same area before? What other monitoring facilities are in place? The most interesting part is how to install a Raspberry Shake in alpine conditions for long-term use. Please slightly expand your descriptions here. How is the Raspberry Shake fixed to the ground? Do you put additional shielding against wind or thermal effects? How is time synchronized? Photographs showing more details would be of great help – maybe as supplemental material.

**Reply: There are currently some seismic monitoring and geophysical studies ongoing, but the results are not available to us. In section 3 we have now better described the installation and included more pictures about the installation sites in the Supplementary Information.**

Section X

At this part of the manuscript I am missing a section about the performance of the network. What's the amount of data retrieved? Where there any power downs? Why so? What's the quality of the data? Any gaps? What's the noise level? I would certainly like to see some PPSD/PDFs showing the noise levels over one year. Is the relative timing good enough? Any other issues due to environmental conditions?

**Reply: These are all very good points. We have now included PSD/PDFs of the RS for representative weeks in different seasons against a broadband station (CH.FIESA). Details are found in section 4.1.**

Section 4

- Generally, this section is very simple and should be greatly supported by a description of the methods you used. My general suggestions is: instead of elaborating on individual events that after all are hard to interpret, I suggest to make use of the long observation time and large number of detected events. Are there some patterns to be seen? How do amplitudes distribute? Is it in fact possible to locate the events? Is all of this actually possible with your dataset? If yes – please show! If no – try to discuss limitations.

**Reply: This is part of our current research and beyond the scope of this paper**

- The detection threshold of the network will rely on the average noise levels throughout the year. Can you give an estimate of what's the smallest ground motion you can reliably detect? This would be of relevance for detecting both earthquakes and rockfalls.

**Reply: The smallest theoretical ground motion detectable is in the order of 0.14x1e-6 m/sec (as per tech. spec.) This is confirmed in our experience, as the background noise levels reported (see Figure 2) are in line with the self-noise of the instrument.**

What are the settings for the STA/LTA detector?

- How did you identify the 250 events as rockfalls? This is a very crucial task for any monitoring so please be more specific on how you identify those. Did you also use an STA/LTA? On how many stations do you see those events? How many of the events are confirmed by the webcam? Was it a visual inspection of the seismic data or some automatic processing? If the latter, please provide parameters or criteria.

- How many events did you detect in total, how many of those are of "unknown" type? How did you e.g. identify rainfall (example in supplemental material) in your seismic data? How the helicopter?

**Reply: Currently, the 250 events as well as the other events were identified with visual inspection. The automatic STA/LTA detection procedure is being currently developed after the preliminary assessment on the RS performances in the field.**

- Figure S3 is quite an interesting observation! Obviously one needs to trust the detections (thus, see comment above), but certainly this is a nice observation that should be moved into the main body of the manuscript.

**Reply: see reply above. Also the analysis of the dataset will be part of future publications.**

- Unfortunately the webcam images (Figure 3) are very hard to interpret (especially when printed on paper). Are these the best examples you can show? What's your experience with the 10 minute sampling rate? Is it sufficient? What would be the effort to increase it?

**Reply: We have now included in the supplementary Information the pictures at a higher resolution and zooms on the portions where rockfalls were detected.**

- I am missing a paragraph dedicated to locating the detected rockfalls. After all this is one of the most important tasks for monitoring purposes. Did you attempt to locate the events? How could it be done? How well does it work? Are there specific drawbacks? - You try to qualitatively interpret some of the amplitude readings for individual events. In general it would be interesting if the recorded amplitudes relate to e.g. distance or ground type underneath the stations or between the stations and the rockfalls. Do you see any general behavior? With 250 rockfalls you should have quite a nice database to dig into this. Is there maybe an effect due to ground-coupling of the sensors?

- You show several examples of seismically recorded rockfalls, yet your description of those events is very basic. In fact, it is no surprise that you are able to detect rockfalls in the close vicinity with the Raspberry Shake. Yet, already from looking your examples it is somewhat obvious (and known from other studies) that rockfall seismic signals are quite complex and relating them to rockfall mechanisms in real field applications is not necessarily straight forward. The real challenge of any monitoring system lies in safely classifying the rockfall events and to discriminate them from all the wealth of other seismic signatures (e.g. anthropogenic) in your data. This starts at classifying seismic signal as "rockfalls" and continues into scaling relations between seismic observables (e.g. amplitude, frequencies, energy, . . .) and rockfall parameters (volume, runout, mechanism, . . .). I think there is interesting work in that direction to be done with this dataset, but it's likely out of scope for the current manuscript. You may decide to work into that direction, but for the moment it would maybe be already sufficient if you add a section that discusses these challenges.

**Reply: Most of the information required in the points above here is part of our current research activity. This will be the focus of future publications.**

————————————————————

Specific remarks/typos/grammar:

Page 1, Line 16: remove "investigations"

**Reply: done**

Page 1, Line 24: remove "of"

**Reply: done**

Page 1, Line 30: what do you mean by arbitrary information?

**Reply: was unclear, we removed.**

Page 2, Line 3: replace "and" with "to"

**Reply: done**

Page 2, Line 5: replace "Despite" with "However"

**Reply: removed**

Page 2, Line 26: ...from the seismic signal IT is possible . . .

**Reply: done**

Page 3, Line 1: Explain IoT

**Reply: removed**

Page 3, Lines 3ff: state somewhere that the sensors measure ground velocity

**Reply: done**

Page 3, Line 18: remove "to"

**Reply: done**

Page 4, Line 1: change to: . . . isolation from different external effects (... )

**Reply: removed**

Page 4, Line 6: what is a Winston Wave Server? What is it good for?

**Reply: we added details.**

Page 4, Line 24: replace "lower" with "shallower"

**Reply: done**

Page 5, Line 5: Figures 3 ! Figure 3

**Reply: done**

Page 5, Line 9: add reference to Figure

**Reply: done**

Page 5, Line 14: replace "presents" with "measures" or "records"

**Reply: done**

Page 5, Line 29: replace "coeval" with "consecutive"

**Reply: done**

Page 5, Lines 26ff: This part should be moved upwards, since it contains the detection/

verification of events

**Reply: removed**

Page 5, Line 31: Give examples for "other processes occuring in the subsurface"

**Reply: we mentioned now creep and stick-slip behavior, and cite Poli et al., 2017**

Page 6, Line 1: Change to "Apart from rock fall phenomena . . . signals associated to

other environmental processes (such as. . .)

**Reply: done**

Page 6, Lines 11ff: You should only summarize your findings. The advertisement for low cost solutions should be moved to the introduction section.

**Reply: done**

————————————————————

Figure 1:

- Add arrow indicating the line-of-sight of the webcam

**Reply: this is now in the Supplementary Information.**

- DSGSD is not explained

**Reply: deleted in the revised version.**

- Show inside installation of RS-3 as well

**Reply: this is now in the Supplementary Information.**

Figure 2:

- Convert Lat/Lon to distance or add distance to the plot titles.

**Reply: added distance**

- Caption: b) Your plots show examples of detected events and maybe estimate a lower

threshold. There is no information about the real probability of detecting an event.

**Reply: the PDF figures added help now in defining the real probability of detecting events vs. a nearby broadband stations**

Figure 3:

- Any chance to improve the contrast/visibility of the webcam pictures? It is really hard to see any changes in the before/after pictures. Why not show some examples from daylight period? These should show a clear before/after signature?

**Reply: we moved the pictures in the Supplemetary information to increase the view before and after rock falls.**

- Caption: august ! August

**Reply: modified**

Figure 4:

- Please describe this Figure in more detail in the manuscript text. You show the waveforms in great detail. If there is something noteworthy to be seen, please state so.

**Reply: done**

- What's the filter applied to the waveform data?

**Reply: Butterworth band pass filter (0.15-15 Hz)**

———————————————————

Supplemental material:

- The text is mainly a copy of Section 2 and I think it can be dropped. If you like to describe the very specific details of the signal processing inside the Raspberry Shake, you could do that here (and not in Section 2).

**Reply: modified**

- Figures S1 and S2 are copied from the producer spec sheet. You could just refer to that sheet. If you like to keep them, please indicate the source.

**Reply: see reply to reviewer 1**

- If you have any, here would be the perfect place to put more examples (waveforms + webcam pictures) of rockfalls (and maybe other sources: cable car, mountaineers, ...)

**Reply: this will be included in future papers.**

---

## Author Response (AR2)

Associate Editor Decision: Publish subject to minor revisions (review by editor) (21 Nov 2018) by Fabian Walter

Comments to the Author:

Dear authors,

thank you for resubmitting your manuscript. I appreciate that you included more technical details on the instruments as the referees had requested. At this point, your manuscript requires minor revisions before reaching publication quality of a short note. Note that this is based on my own judgment as I deemed it unnecessary to contact the referees with the revised manuscript again.

I do ask you to respond to my comments below. Moreover, although the text is clearly written and easy to follow, there are several typographical and grammar mistakes, which is why I ask you to give it a thorough final read, ideally from a native English speaker.

Best,

Fabian Walter.

**Dear Fabian, thanks for your decision to accept our manuscript pending minor revision. Please find below our point-by-point replies to your final comments. Best regards, Andrea**

1. Citations: At various points in the manuscript you refer to typical seismic signatures of rock-fall and earth surface processes (e.g. Line 5 on Page 2 and Line 9 on Page 6). I suggest you give more references, as other research groups have specifically worked on compilations of such seismic signals:

Vouillamoz, N., Rothmund, S., & Joswig, M. (2018). Characterizing the complexity of microseismic signals at slow-moving clay-rich debris slides: the Super-Sauze (southeastern France) and Pechgrabe (Upper Austria) case studies. Earth Surface Dynamics, 6(2).

Hibert, C., Malet, J. P., Bourrier, F., Provost, F., Berger, F., Bornemann, P., ... & Mermin, E. (2017). Single-block rockfall dynamics inferred from seismic signal analysis. Earth Surface Dynamics, 5(2), 283-292.

Provost, F., Malet, J. P., Hibert, C., Helmstetter, A., Radiguet, M., Amitrano, D., ... & Lebourg, T. (2018). Towards a standard typology of endogenous landslide seismic sources. Earth Surface Dynamics, 6(4), 1059-1088.

**Reply: We have included the suggested references, thanks.**

2. PSD PDF's in Figure 2: For the RS sensors you show very long periods (beyond a few seconds), where the sensor is no longer sensitive. This should be labeled at least. Moreover, there is a clear branching for RS-

3. I assume that this is the cable car operation (which you mention in the text), which could significantly affect detectability of weak rock fall signals at this station. In the worst case this may cause spurious diurnal activity cycles interfering with natural process cycles (Figure S5). Please comment on this and the general comparison with FIESA. Also, please include a reference for the Peterson noise models mentioned in the caption of Figure 2.

**Reply: We have now better specified this point in the revised version. Figure S5 is done by considering events detected at least by the stations RS-1 and RS-2, and in the best case also by RS-3. Thus, there is no bias due to the cable-car operations in our preliminary interpretations. We have now also included a reference for the Peterson's curves.**

Page 2, Line 2: The last sentence is unclear.

**Reply: We have now deleted this sentence.**

Page 3, Line 25-26: touch by hands greater than 50 milimeters is unclear.

**Reply: This is the standard specification on the IP levels. We have now removed the unclear text.**

Figure 5: Specify Butterworth filter (2nd order?)

**Reply: Yes is a 2$^{nd}$ order Butterworth filter, this is now specified.**

Figure S4, left image: Please annotate or remove as it is hard to understand what you are showing (wheelbarrow?).

**Reply: We have now modified the figure and added more info in the caption.**

Figures S9.?: The zooms are very helpful, however, even within them it is hard to make out differences. Please add arrows that guide the reader's eyes.

**Reply: We have now added the arrows, thanks.**

Details on NTP performancs: This is useful information. The plots, however, are enigmatic for someone who sees them for the first time. I understand that this is standard output and cannot be modified easily, so please provide more explanations.

**Reply: We have now added a reference for reading the plots in the figure caption.**

**Short Communication: Monitoring rock falls with the Raspberry Shakes**

Andrea Manconi[1*], Velio Coviello[2], Maud Galletti[1], Reto Seifert[1]

[revised manuscript text omitted]

**4 Results**

**4.1 Monitoring performance**

RS-1 and RS-2 stations, both installed on the ground surface at elevations >2,000 m a.s.l. in an alpine environment, provided continuous record of seismic data since the installation without any site intervention in the 1-year monitoring period presented here. Air temperatures in this period ranged from -20 degrees in winter to +25 degrees in summer, and snow cover up to 3 meters was recorded at the RS-2 location and around 1.5 meters at RS-1 location between January and March 2018. This confirms that the enclosure we have deployed was sufficient to protect the Raspberry Pi components against alpine environmental conditions. We reported only very limited data loss (in total less than 5 minutes records over 1 year,-) at the stations RS-1 and RS-2, associated to planned system restarts after configuration changes (performed through remote access). At the RS-3 location instead, the data loss was more consistent (in total one week of data loss), due to power outage at the cable car station during a period of planned maintenance. However, the problem was unrelated to the RS-3 system itself, which started again to properly record data without intervention when the power was set back to normal. Data transfer through the cellular network links (RS-1 and RS-2) also worked smoothly during 1-year period. The results of systematic ping-tests (20 ICMP echo pings of 56 bytes every 300 seconds) show an average response time below 100 milliseconds. No remarkable network outage is reported during the period of observation (see also Supplementary Information), ensuring thus continuity for potential near-real-time analyses, as well as for the NTP service synchronization. Estimated timing quality is thus in the order of ±0.02 seconds (1 sample) or better. The current network density (3 stations with inter-station distance of about 1 km) is sufficient for detection and validation of the seismic signals but probably not enough to achieve accurate source locations. These inaccuracies can be further enhanced by time synchronization issues between the stations due to the use of NTP services; however, expected timing errors are in the order (or smaller) of the biases due to incorrect velocity models or imprecise phase picking (Lacroix and Helmstetter, 2011)(Anthony et al., 2018; Lacroix and Helmstetter, 2011).

We investigated the quality of the seismic data acquired, by comparing the background noise (McNamara and Buland, 2004) of our three RS against a reference broadband seismic station (CH.FIESA, managed by the Swiss Seismological Service (SED), see details at http://stations.seismo.ethz.ch) located at about 5 km distance from the RS-1 station (Figure 2). The results show

5   that the RS stations performed within the expected boundaries for such low-cost sensors (see also nominal instrumental noise levels in the Supplementary Information). As expected, the main difference between the CH.FIESA and our stations is the performance for long period signals (>10 seconds), due to bandwidth limitation of the RS sensors. We note also that during winter, performances for short period signals (0.1-1 seconds) are comparable to CH.FIESA, while in summer are still within

10  the noise model boundaries (Peterson, 1993) but slightly worse. This is probably because in winter the snow cover (maximum during the observation period 3 meters at RS-1 and 1.5 meters at RS-2) protected the sensors (which are installed at the surface) against surficial noise sources. Moreover, during winter the glacial environment is relatively quiet compared with the spring and summer periods, when during the day surface water runoff, as well as glacier flows, are very active and may affect the background noise levels. In addition, anthropic disturbances in this region are stronger during summer periods due to the large

15  number of tourists visiting the Great Aletsch area. The data acquired at RS-3 systematically suffered from a higher noise levels (see the clear PSD/PDF branching in Figure 2) during the cable car operational time period (between 8am and 4.30 pm local time), while during evenings and nights the background noise levels are similar to RS-1 and RS-2 stations.

20  **4.2 Earthquakes**

In a monitoring scenario where the main interest is to detect rock falls, recognition of earthquake events in the seismic traces is very important for two main reasons: (i) ground shaking due to local earthquakes (distances <100 km) can cause rock falls (e.g., Romeo et al., 2017), thus their identification is important to properly study the triggering factors affecting the rock slope degradation; (ii) the signals associated to distant events, such as regional earthquakes (distance >100 km) and teleseisms

25  (distance >1000 km) have characteristics that might be similar (in terms of amplitudes and durations) with the signals caused by mass wasting phenomena  (Dammeier et al., 2011; Helmstetter and Garambois, 2010; Manconi et al., 2016; Provost et al., 2018), and thus introduce a bias in the aimed rock fall catalogue. In order to test the performance of our local RS network, we selected seismic events from

30  the catalogue provided by USGS (NEIC, see catalogue in the Supplementary Information, table S1), considering crustal events at depths shallower than 50 km, magnitudes larger than M2.5, and occurred at distances up to 15,000 km from our study area within 1-year time period (May 19, 2017 and May 19, 2018). We found that 47 out of the 64 selected earthquake events (~73%) were clearly visible in the waveform recorded by the RS-1 (Figure 3). As expected, the detectable magnitude as well as the

signal amplitude scales with the distance from the seismic event's source. From the waveforms (Figure 3 a-d) it is possible to recognize the main differences in terms of amplitudes, duration and signal characteristics for different events.

**4.3 Rock fall signals**

About 250 rock fall events have been visually identified in the seismic traces (recorded at least in two stations) during period

[revised manuscript text omitted]

**Figures**

[Figure]

**Figure 1: (left) Map of the area of investigation with indication of the location of the three RS seismic station installed starting from May 2017. (a-c) Pictures of the RS installation (a, RS-1; b, RS-2; c, RS-3). Continuous records of seismic signals on the 3 stations are available since beginning of July 2017.**

[Figure]

**Figure 2: Comparison of background noise levels between a broadband station (CH.FIESA) and the Raspberry Shake stations (RS-1, RS-2, and RS-3) installed in the Aletsch region for 1-year. Probability Density Functions (PDF) of the Power Spectral Densities (PSDs) were computed by stacking windows of 10 minutes in two reference weeks, one in winter (top row, 01-08 March, 2017) and one in summer (bottom row, 01-08 August, 2017). The black lines represent  high and low reference noise models. The broadband station CH.FIESA managed by the Swiss Seismological Service is installed in the Aletsch region about 5 km away from the RS network. Branching of PSD/PDF at RS-3 is caused by diurnal operations of the cable car.**

[Figure]

M3< 🟢 4 🟡 5 🟠 6 🔴 7 🟣 >M8

[Figure]

**Figure 3: Performance of the RS-1 station in recording earthquakes. (top) Spatial distribution of earthquake events identified in the RS-1 waveforms out of a catalogue of 64 earthquakes occurred within 1-year time period at distances up to 15,000 km. (a-d)**
5 **Examples of seismic signal recorded by RS-1 associated to earthquakes of different magnitudes and occurred at increasing distances from the monitoring station. Signals are band-pass filtered (Butterworth, 2nd order) between 0.5 and 15 Hz.**

[Figure]

**Figure 4: Selection of signals associated to rock fall events. Signals are band-pass filtered (Butterworth, 2ⁿᵈ order) between 0.5 and 15 Hz Time is in UTC. Note the large noise level at the station RS-3 caused by the cable car operations (see also section 4 for more details).**

[Figure]

[Figure]

Figure 5: Detail of a rock fall event occurred on July 27, 2017 around 15:37 UTC. (Top) Seismic signal is clearly visible at the three RS stations. Note the differences in amplitudes and phases. (Bottom) Three snapshots with 10 minutes baseline acquired by the webcam. The rock fall event is clearly visible (white circle). Future work will jointly exploit seismic and optical images to locate and characterize rock fall events.